# MWCNTs Composites-Based on New Chemically Modified Polysulfone Matrix for Biomedical Applications

**DOI:** 10.3390/nano12091502

**Published:** 2022-04-28

**Authors:** Simona Luminita Nica, Mirela-Fernanda Zaltariov, Daniela Pamfil, Alexandra Bargan, Daniela Rusu, Delia Mihaela Raţă, Constantin Găină, Leonard Ionut Atanase

**Affiliations:** 1Petru Poni Institute of Macromolecular Chemistry, 41A Grigore GhicaVoda Alley, 700487 Iasi, Romania; zaltariov.mirela@icmpp.ro (M.-F.Z.); pamfil.daniela@icmpp.ro (D.P.); anistor@icmpp.ro (A.B.); rusu.daniela@icmpp.ro (D.R.); gcost@icmpp.ro (C.G.); 2Faculty of Medical Dentistry, Apollonia University of Iasi, Pacurari Street, No. 11, 700511 Iasi, Romania; delia.rata@univapollonia.ro

**Keywords:** multiwall carbon nanotubes, modified polysulfone, polyvinyl alcohol, composite, morphology, water sorption, blood compatibility

## Abstract

Polyvinyl alcohol (PVA) is a non-toxic biosynthetic polymer. Due to the hydrophilic properties of the PVA, its utilization is an easy tool to modify the properties of materials inducing increased hydrophilicity, which can be noticed in the surface properties of the materials, such as wettability. Based on this motivation, we proposed to obtain high-performance composite materials by a facile synthetic method that involves the cross-linking process of polyvinyl alcohol (PVA) with and aldehyde-functionalized polysulfone(mPSF) precursor, prior to incorporation of modified MWCNTs with hydrophilic groups, thus ensuring a high compatibility between the polymeric and the filler components. Materials prepared in this way have been compared with those based on polyvinyl alcohol and same fillers (mMWCNTs) in order to establish the influence of the polymeric matrix on the composites properties. The amount of mMWCNTs varied in both polymeric matrices between 0.5 and 5 wt%. Fourier transformed infrared with attenuated total reflectance spectroscopy (FTIR-ATR) was employed to confirm the changes noted in the PVA, mPSF and their composites. Hemolysis degree was investigated in correlation with the material structural features. Homogenous distribution of mMWCNTs in all the composite materials has been confirmed by scanning electron microscopy. The hydrophilicity of both composite systems, estimated by the contact angle method, was influenced by the presence of the filler amount mMWCNTs in both matrices (PVA and mPSF). Our work demonstrates that mPSF/mMWCNTs and PVA/mMWCNTs composite could be used as water purification or blood-filtration materials.

## 1. Introduction

Development of materials as porous polymeric membranes to be used in some particular applications (blood filtration, water purification, etc.) rely on different preparation techniques [1,2], which are well-controlled. It is important to know how the targeted liquid will act on the surface of the obtained material. Recently, the fabrication processes of materials with controlled wettability have been widely exploited [3,4]. The choice in adjustment of the polymers surface in terms of wettability [5,6] is often employed. To this is added equally the design of a specific architecture possessing adequate functionalities and a good permeability, essential for blood filtration membranes [7,8]. Different physical (plasma treatment, physical coating, immobilization of nanoparticles) and chemical (reaction between the newly introduced surface functional groups, photoinitiation, redox) approaches for surface modification have been reported [9]. Relevant work describing the implementation of a new method to prepare composites with a complex structure was reported by Donchak and co-workers [10], who have chosen to introduce amino groups on the surface of MWCNTs, which were later reacted with the carboxylic groups of the polymeric matrix, improving the tensile strength and the elasticity of the composites.

In another approach, Melnyk and co-workers [11] described an optimized procedure to prepare cross-linked hydrogel membranes based on poly(vinylpyrrolidone)-graft-poly(2-hydroxyethylmethacrylate) designed for drug delivery systems.

One of most representative class of materials recognized for their specific applications in membranes for blood purification is polysulfones (PSFs) [12], even if they are hydrophobic and manifest a lack of compatibility with blood [13,14,15,16,17]. To improve their biocompatibility, especially preventing the occurrence of coagulation cascade and thrombosis phenomena [18,19], different approaches had been investigated. One of them is functionalization of polysulfones with quaternary ammonium groups to obtain hemocompatible polymeric membranes with higher porosity and hydrophilicity [20]. An increased biocompatibility with blood was reported by Qi et al. [21], who incorporated antioxidant compounds (resveratrol). The reported membranes showed a high efficiency in blood filtration (about 90.33% urea, 89.50% creatinine, 74.60% lysozyme, and 90.47% BSA) and in reducing of the oxidative stress. In another approach, Zhong et al. [22] reported polysulfone-block-poly(ethylene glycol) (PSF-b-PEG) ultrafiltration membranes for hemodialysis by non-solvent induced phase separation (NIPS) method in the absence of any additives. Another synthetic polymer that received attention as possible blood-filtration material is polyvinyl alcohol (PVA) [23,24], which possesses required characteristics for biomedical applications such as mechanical strength, biocompatibility, and non-toxicity [25].

The embedding of different fillers [26] in polymer matrices to obtain composite membranes is another way to improve separation properties of membranes. Among them, carbon nanotubes (CNTs), a class of materials with tubular structures, received great interest. In this respect, composites based on multiwall carbon nanotubes (MWCNTs) seem to have a multitude of practical uses [27,28,29]. Despite different opinions of the researcher regarding the toxicity of CNTs, the composite membranes based on MWCNTs proved their utility in medicine in cancer treatment [30], bio-sensing [31], hyperthermia induction [32], antibacterial therapy [33], bone tissue engineering [34], and blood-contacting devices [35]; in respirators, allowing the free exchange of air and CO_2_ and preventing the transmission of viruses and bacteria; in drug delivery of small molecules, proteins, or nucleic acids; and in protein purification and selective permeation [36]. MWCNTs have a hydrophobic nature. Surface charge influences the biological performance [37], so their applications as biomaterials demand surface functionalization. As a consequence, the solubility in solvents and their biocompatibility are improved [38]. Detailed information regarding biocompatibility of functionalized MWCNTs (denoted here mMWCNTs) can be found in the literature [39]. Much less discussed and evaluated is the hemocompatibility of mMWCNTs. These represent a mandatory requirement for materials, as tests referring to hemocompatibility can prove either their failure or their success in biomedical applications. The surrounding tissue is perturbed by the implanted “foreign” biomaterial. Therefore, implanted blood-contacting devices may produce thrombo-inflammation. It has been demonstrated that materials having hydrophilic surfaces are resistant to blood protein adsorption [40]. The advantage conferred by the surface functionalization of MWCNTs shows a decrease intoxicity and improvement of antifouling properties [41]. Literature data revealed the utilization of carbon nanotubes in microfluidic devices with higher efficiency (80%) in plasma separation from blood. The intrinsic porosity of CNTs of approximately 93% resulted from the inter-tubular distances lower than 90 nm [42]. Electroanalytical devices based on multi-walled carbon nanotubes also proved to be efficient in rapid detection and L-tyrosine monitorization in tyrosinemia diagnosis, with a high degree of reusability (97%) [43] and in blood glucose in humans with a higher selectivity than the grapheme-based biosensors, with MWCNTs being a superior electrode material [44].

CNTs are also used as carriers in drug delivery systems. Functionalized MWCNTs conjugated with inorganic nanoparticles (iron oxide) or biological molecules (biotin) have been investigated for doxorubicin loading and release. CNTs can be intravenously administered, showing some cytotoxic and genotoxic side effects, producing increased intracellular ROS species, DNA damage, and some inflammation at concentrations up to 50 μg/mL [45]. Pulmonary inhalation of MWCNTs has also altered rat heart rate variation, affecting the nervous system [46]. In-depth studies have shown, however, that a specific functionalization of MWCNTs was beneficial in permeation of the blood-brain barrier. The cell viability tests proved that these mMWCNTs were safe; no cytotoxic effects were reported. On the contrary, they showed the ability to function as central nervous system-targeting drug release materials [47]. The main advantage is related to their unique and controlled structure and intrinsic porosity, which can be promoted in preparation of stable drug delivery systems and blood devices. Another strong point is that MWCNTs can be easily functionalized and compatibilized with different functional groups/substrates, so they are accessible in different applications. In addition, their embedding in polymeric matrices proved to increase the thermal stability, flexibility, and stability of materials.

The advantage of the composites used as membrane separation materials in blood purification processes relies on their high permeability, easy fabrication, good biocompatibility, fast adsorption rate, and high adsorption capacity of nanomaterials [48].

Based on the literature background, the current paper aims to obtain and characterize new composite materials for applications in water purification membranes or blood-contacting materials. Our strategy is based on original aspects that arise from the chemical modification of the PSF (mPSF) by introduction of PVA segments as side chains. With this procedure, we expect to improve the wettability and porosity of the prepared materials. Moreover, the properties of these membranes were further enhanced by the addition of multiwall carbon nanotubes that were chemically functionalized with -OH groups. For comparison purposes, the PVA/mMWCNTs nanocomposite system was also analyzed. All tested composite materials were evaluated in terms of wettability, water sorption properties, and hemocompatibility. These new materials open new perspectives for improving the performance of current membranes used for water purification membranes or blood filtration.

## 2. Experimental

### 2.1. Materials

Commercial multi-walled carbon nanotubes (MWCNTs, 95% purity, PD15L5-20) with an outer diameter of 15 ± 5 nm and a length of 5–20 µm were purchased from NanoLABInc (Waltham, MA, USA) and used as received.

Polysulfone (Udel 3500P, Aldrich, Schnelldorf, Germany), *p*-formaldehyde (95%, Aldrich, Schnelldorf, Germany), chlorotrimethylsilane (99%, Aldrich, Schnelldorf, Germany), tin (IV)chloride (99%, Aldrich, Schnelldorf, Germany), poly(vinyl alcohol) (PVA) (viscosity-average molecular weight 77,000–79,000 g/mol, DH = 98%, Aldrich, Schnelldorf, Germany), *p*-toluenesulfonic acid (pTSA, 97%, Aldrich, Schnelldorf, Germany), *N*-methyl-2-pyrrolidone (NMP, Aldrich, Schnelldorf, Germany), chloroform, and sodium bicarbonate (Chemical Company, Iasi, Romania) were used as received.

### 2.2. MWCNTs Modification and Composite Synthesis

To produce MWCNTs functionalized with hydroxyl groups (mMWCNTs), pristine MWCNTs were reacted in a H_2_O_2_ solution, according to a reported method [49].

Chloromethylated polysulfone (ClMePSF) was prepared according to existing literature data [36,50] through an approach described below. Subsequently, aldehyde-functionalized polymer (APSF) was obtained by direct-oxidation of ClMePSF in DMSO, in the presence of NaHCO_3_, according to [51] (Figure 1).

Chloromethylated polysulfone (PSF-ClMe): In a typical reaction, polysulfone (5 g) was dissolved in chloroform (200 mL) in a round bottom flask equipped with a magnetic stirrer, reflux condenser, and thermometer. The solution was heated at 50 °C under nitrogen atmosphere and then *p*-formaldehide (3 g), chloromethylsilane (14 mL), and thin (IV) chloride (0.6 g) were added to the polymer solution. After 50 h at 50 °C, the mixture was poured into methanol and the solid product washed well with methanol and dried at 60–65 °C. The degree of substitution of –CH_2_Cl groups was evaluated using ^1^H-NMR spectroscopy and was estimated at around 1.2. Subsequently, aldehyde-functionalized polymer (APSF) was obtained by direct-oxidation of PSF-ClMe in DMSO, in the presence of NaHCO_3_, according to [51] (Figure 1). A typical synthesis, PSF-ClMe (5 g) was dissolved in DMSO (120 mL), and sodium bicarbonate (2.1 g) was added. The reaction mixture was maintained at 130 °C for 12 h, and then poured into water.

By acetalization of PVA with APSF in NMP, mPSF was obtained. The PSF-PVA mixture was defined as mPSF, and the mass ratio of APSF:PVA was 1:1. APSF (0.3 g) and PVA (0.3 g) were dissolved in NMP individually to form a solution of 6.5%. The two solutions were then mixed together at room temperature, and, subsequently, pTSA was added and heated at 60 °C for 8 h. The mPSF membranes were prepared by solution casting method as follows: To a solution of mPSF (0.6 g) in NMP (8 mL), an appropriate amount of mMWCNTs was added to obtained mPSF/mMWCNTs composition of 0.1, 1, 3, and 5% (Figure 1). The mixtures were sonicated at 60 °C for 30 min to form a black viscous homogeneous solution, then were cast on the glass plates and dried at 80 °C for 12 h. The obtained mPSF membranes were then thoroughly washed with water and dried at 80 °C for other 8 h.

### 2.3. Methods

Attenuated total reflectance–Fourier transform infrared spectroscopy (ATR–FTIR) measurements were carried out on a Bruker Vertex 70 spectrometer (Bruker Optics, Ettlingen, Germany) equipped with a ZnSe crystal. The spectra were recorded in ATR (attenuated total reflectance) mode in the 4000–600 ${\mathrm{cm}}^{-1}$ range, with a resolution of 4 ${\mathrm{cm}}^{-1}$ and 32 scans, at room temperature.

The static contact angle experiments were performed on KSV CAM 200 goniometer at room temperature (25 °C). The sessile-drop method involved the utilization of two test liquids (water and ethylene glycol) that were handled with separate testing syringes to avoid cross-contamination. A liquid droplet of 1 μL was formed at the end of the syringe and carefully deposited onto the surface of prepared film samples, and then the image of droplet was taken by a coupled device camera in order to measure the contact angle. Tests were repeated three times on different surfaces.

To obtain the surface tension parameters, the Fowkes method/model [52] was employed as shown in Equations (1) and (2):(1)$$\frac{\left(1+\cos\theta \right)\cdot \gamma_{lv}}{2}=\sqrt{\gamma_{sv}^{d}\cdot \gamma_{lv}^{d}}+\sqrt{\gamma_{sv}^{p}\cdot \gamma_{lv}^{p}}$$
(2)$$\gamma_{sv}=\;\gamma_{sv}^{d}+\gamma_{sv}^{p}$$
where θ is the contact angle formed by a liquid on the solid surface samples and *γ^p^_lv_*, *γ^d^_lv_*, *γ^p^_sv_*, and *γ^d^_sv_* are the polar and dispersed components of the liquid and solid phases, respectively.

Solid-liquid interfacial tension ($\gamma_{sl}$) is defined in Equation (3):(3)$$\gamma_{sl\;}={\left(\sqrt{\gamma_{l}^{p}}-\sqrt{\gamma_{s}^{p}}\right)}^{2}+{\left(\sqrt{\gamma_{l}^{d}}-\sqrt{\gamma_{s}^{d}}\right)}^{2}$$

The spreading coefficient, $S_{c}$, is defined by Equation (4):(4)$$S_{c}=\gamma_{s}-\gamma_{sl}-\gamma_{l}$$

Water adsorption/desorption isotherms of all composite systems were registered using a Dynamic Vapor Sorption (DVS)Analyzer (IGAsorp, Hiden Analytical, Warrington, UK) equipped with an ultrasensitive microbalance to measure the weight changes when the relative humidity is varied. Before sorption experiments, the samples were dried at 25 °C in flowing nitrogen (250 mL/min) until equilibrium at RH < 1%. The relative humidity (RH) was then slowly increased from 0 to 90%, in 10% humidity steps, with every step having a pre-established equilibrium time between 40 and 60 min. The sorption equilibrium was obtained for each step. After that, the RH was decreased and desorption isotherms were registered.

The Brunauer–Emmett–Teller (BET) kinetic model [53] was applied to calculate the surface area and monolayer of the synthesized materials according to Equation (5):(5)$$W=\frac{W_{m}\cdot C\cdot \mathrm{RH}}{\left(1-\mathrm{RH}\right)\cdot \left(1-\mathrm{RH}+C\cdot \mathrm{RH}\right)}$$
where W is the weight of sorbed water, $W_{m}$ is weight of water forming a monolayer, C is the sorption constant, and RH is the relative humidity.

The surface morphology of the materials was examined by using a scanning electron microscope (SEM) type Quanta 200 (FEI), operating at 20 kV with secondary electrons in low vacuum mode, the magnification being indicated on each micrograph. The SEM micrographs were performed on uncoated samples fixed on aluminum stubs.

### 2.4. Hemolysis Assay

The compatibility of the obtained films with human red blood cells (RBCs) and their potential effects on erythrocyte lysis were evaluated by the hemolysis test using a spectrophotometric method adapted from Chhatri et al. [54] and Rata et al. [55].To perform this test, a healthy, non-smoking volunteer was selected from whom 5 mL of blood was collected, after prior institutional ethical approval and adequate informed consent. The blood was centrifuged at 3000 rpm for 5 min and washed several times with normal saline solution (0.9% NaCl) to remove plasma and to obtain erythrocytes. The purified erythrocytes were re-suspended in normal saline to obtain 25 mL of erythrocyte suspension. All the investigated films had a spherical shape with a diameter of 5 mm and were transferred into Eppendorf tubes, over which 0.5 mL of erythrocytes suspension and 0.5 mL of normal saline solution were added. Positive control samples, with 100% lysis, were prepared by adding 0.5 mL of 2% Triton X-100 solution over 0.5 mL of erythrocyte suspension. Negative control samples, with 0% lysis, were prepared by adding 0.5 mL of saline solution over 0.5 mL of erythrocyte suspension. The samples were incubated at 37 °C for 3 h and gently shaken once every 30 min to ensure continuous contact between the films and the blood. After the incubation time, the samples were centrifuged at 3000 rpm for 5 min, and 100 μL of supernatant was incubated for 30 min at room temperature to allow oxidation of hemoglobin. The absorbance of oxyhemoglobin in supernatants was measured spectrophotometrically at 540 nm using a Nanodrop One UV-Vis Spectrophotometer. All samples were analyzed in triplicate.

The percentage of hemolysis was calculated using Equation (6):(6)$$\mathrm{Hemolysis}\;\left(\%\right)=\frac{{\mathrm{Abs}}_{\mathrm{sample}}-{\mathrm{Abs}}_{\mathrm{negative}\;\mathrm{control}}}{{\mathrm{Abs}}_{\mathrm{pozitive}\;\mathrm{control}}-{\mathrm{Abs}}_{\mathrm{negative}\;\mathrm{control}}}\times 100$$

Membrane materials were classified into three categories based on their hemolytic index as follows: (1) hemolytic materials which show hemolysis (%) > 5%; (2) slightly hemolytic materials with hemolysis (%) between 2% and 5%; (3) non-hemolytic materials with hemolysis (%) < 2% (American Society for Testing and Materials 2000) [56].

## 3. Results and Discussion

### 3.1. ATR–FTIR Characterization of Polymeric Matrices

The IR data of the polymeric matrices and composites are summarized in Table 1.

The IR absorptions of PVA and mPSF-based matrices in the 4000–2500 ${\mathrm{cm}}^{-1}$ region are generally due to O-H and C-H vibrations. The O-H stretching produces a broad band at about 3350 ${\mathrm{cm}}^{-1}$ and is the result of the presence of intermolecular and intramolecular hydrogen bonding. The precise position of the O-H band is dependent on the strength of the hydrogen bonds within the matrix (Figure 2a). The band is blue shifted at 3340 ${\mathrm{cm}}^{-1}$ in the spectrum of mPSF matrix as a result of the H-bond interactions between PVA and APSF.

The bands at 2920 ${\mathrm{cm}}^{-1}$ and 2856 ${\mathrm{cm}}^{-1}$ are assigned to the asymmetric and symmetric C-H stretching vibrations within PVA, APSF, and mPSF matrices, whereas the bands at 3100 ${\mathrm{cm}}^{-1}$, 3070 ${\mathrm{cm}}^{-1}$, and 3030 ${\mathrm{cm}}^{-1}$ are due to the aromatic C-H stretching vibrations in the structure of APSF and mPSF (Figure 2a).

In the 1800–600 ${\mathrm{cm}}^{-1}$ spectral region of the polymeric matrices, strong characteristic peaks for PVA at 1733 ${\mathrm{cm}}^{-1}$ and 1670 ${\mathrm{cm}}^{-1}$ are due to the incompletely hydrolyze of PVA (the C=O vibrations from acetate groups) [57], 1696 ${\mathrm{cm}}^{-1}$ characteristic for aldehyde CH=O group in the APSF matrix, as well as other specific bands for C-O stretching vibration band in the structure of poly(vinyl acetate) mPSF matrix 1190–1148 ${\mathrm{cm}}^{-1}$. The complete transformation of the carbonyl CH=O group into the acetal was evidenced by the disappearance of the band at 1696 ${\mathrm{cm}}^{-1}$ in the spectrum of APSF. In addition, acetals have a characteristic strong band at 1106 ${\mathrm{cm}}^{-1}$ due to a C-H deformation vibration being perturbed by the neighboring C-O groups (Figure 2b).

Aromatic C=C stretching vibrations occur in the region 1650–1427 ${\mathrm{cm}}^{-1}$, and the skeletal vibrations due to the conjugation can be seen at 1585 ${\mathrm{cm}}^{-1}$.

The bands observed at 1383–1290 ${\mathrm{cm}}^{-1}$ and 1170–1148 ${\mathrm{cm}}^{-1}$ are due to the asymmetric and symmetric stretching vibrations of the ${\mathrm{SO}}_{2}$ group overlapped with the C-O stretching vibrations.

Other bands refer to the in-plane O-H deformation vibration coupled with C-H/C-O wagging vibration (1440–1260 ${\mathrm{cm}}^{-1}$) and C-H rocking vibrations and aromatic ring, whereas the bands at 850–720 and 700–580 ${\mathrm{cm}}^{-1}$ refer to the =C-CH- in-plane deformation vibration, phenoxy (Ph-O) band, and C-H out-of-plane and ring deformation vibrations [58].

### 3.2. ATR–FTIR Characterization of Composites

The O-H region of the composites reveal an increase in the filler concentration of the O-H groups by increasing of the concentration of mMWCNTs as compared with the polymeric matrix (Figure 3a). The interaction of the mMWCNTs with the polymeric matrix through H bonding is highlighted by the blue shift of the band at 3350 with 30 ${\mathrm{cm}}^{-1}$.

As depicted in Figure 3b, all spectra show two absorption bands at 1733 and 1668 ${\mathrm{cm}}^{-1}$, which are attributed to C=O of acetate groups present in the PVA matrix. For the PVA/mMWCNTs composites system, the absorption due to the carbonyl groups is detected at 1657 ${\mathrm{cm}}^{-1}$, whereas that assigned to the acetate is detected at 1713 ${\mathrm{cm}}^{-1}$, indicating a physical bonding through H-bonds of the functionalized mMWCNTs with the polymeric matrix. More than that, a decreasing in the absorption band at 1657 ${\mathrm{cm}}^{-1}\;$ with an increase of the mMWCNTs concentration can be observed, which indicates that the mMWCNTs have been successfully embedded in the polymeric matrix (Figure 3b). In the 1600–1200 ${\mathrm{cm}}^{-1}$ spectral range of the PVA/mMWCNTs composite, the bands at 1506 ${\mathrm{cm}}^{-1}$, 1435, and 1305 ${\mathrm{cm}}^{-1}$ assigned to the O-H deformation vibration and bonded OH groups indicated the successful incorporation of the MWCNTs (Figure 3c).

In the 1200–600 ${\mathrm{cm}}^{-1}$ spectral range, a new band at 1140 ${\mathrm{cm}}^{-1}\;$ appears in the spectra of mMWCNTs composites, being assigned to the PVA (C-O crystallinity) [59], whereas the bands at 1060 ${\mathrm{cm}}^{-1}$, 984 ${\mathrm{cm}}^{-1}$, and 913 ${\mathrm{cm}}^{-1}$ attributed to the C-O stretching are diminished in the spectra of composites (Figure 3d).

For the mPSF/mMWCNTs composites, the absorption peak assigned to –OH stretching at 3340 ${\mathrm{cm}}^{-1}$ indicated the presence of the hydroxyl groups, whose concentration increases as compared with the matrix (Figure 4a). Characteristic peaks for C=O stretching, C=C stretching, and O-H deformation vibrations are also detected at 1729 ${\mathrm{cm}}^{-1}$, 1713 ${\mathrm{cm}}^{-1}$, and 1654 ${\mathrm{cm}}^{-1}$, respectively (Figure 4b).

In the 1500−600 ${\mathrm{cm}}^{-1}$ range, the mPSF/mMWCNTs composites have very similar absorption bands: 1487 ${\mathrm{cm}}^{-1}$(O-H deformation vibration coupled with C-O and C-H wagging vibration), 1429 ${\mathrm{cm}}^{-1}$ and 1407 ${\mathrm{cm}}^{-1}$ (C-C stretching in aromatic rings), 1373 ${\mathrm{cm}}^{-1}$ (C-SO_2_-C asymmetric stretching), 1298 ${\mathrm{cm}}^{-1}$ (S-O stretching), 1240 ${\mathrm{cm}}^{-1}$ (C-O-C symmetric stretching) (Figure 4c), 1148 ${\mathrm{cm}}^{-1}$ (C-SO_2_-C symmetric stretching), 1104 ${\mathrm{cm}}^{-1}$, 1096 ${\mathrm{cm}}^{-1}$, 1032 ${\mathrm{cm}}^{-1}$, 1012 ${\mathrm{cm}}^{-1}$, and 834 ${\mathrm{cm}}^{-1}$ (aliphatic C-O-C, aromatic C-H bending and rocking), 750–650 ${\mathrm{cm}}^{-1}$ (C-H out-of-plane deformation vibrations) (Figure 4d), proving that after embedding the mMWCNTs in the matrix, the chemical structure of the mPSF did not change [60].

The differences in composition of the PVA and mPSF matrices before and after embedding of mMWCNTs were highlighted by using the method of IR spectral subtraction of the spectra, and the results are indicated in Figure 5. In this case, the resulting subtraction spectrum will correspond to the difference between the spectrum of the initial matrix and the spectrum of the composites. Spectra were processed by using Bruker OPUS 6.5 software. It can be seen that the main differences between the initial matrix and composites consist of the characteristic bands of O-H groups (O-H stretching at about 3300 ${\mathrm{cm}}^{-1}$ and 1643 ${\mathrm{cm}}^{-1}$, O-H deformation vibration at 1510 ${\mathrm{cm}}^{-1}$, 1306 ${\mathrm{cm}}^{-1}$, 930 ${\mathrm{cm}}^{-1}$, and 814 ${\mathrm{cm}}^{-1}$) (Figure 5).

### 3.3. Wettability and Surface Free Energy Analysis

In bio-applications, surface characteristics of polymeric materials are considerably influenced by the hydrophobic/hydrophilic balance. Contact angle analysis was performed in order to determine the surface tension of the pure polymer matrices and of the OH modified MWCNTs (mMWCNTs) composites. In addition to this, the wettability properties of the materials, here under study, affect water sorption and hemocompatibility. Table 2 presents experimental data regarding contact angles measured on the surface of prepared matrices and corresponding composites. The standard deviations (SD) were calculated from the values measured in three different positions on the surface region of each sample material. Values are given in Table 2. For all samples, standard deviations are below 2°. It was observed that majority of the probes present low SD indicating more homogenous surfaces. As a consequence, low local heterogeneities exist. For mPSF/mMWCNTs composites (5 wt%), the increased SD is attributed to an increasing amount of surface inhomogeneity. In this situation, the surface hydroxyl group density decreases [61]. An opposite trend was observed for the last measurement in the system PVA/mMWCNTs.

A lower water contact angle indicates a more hydrophilic material surface. As showed in Table 2, the pure PVA matrix has a relatively hydrophilic behavior (θ < 90°). Because PVA is a hydrophilic polymer, it is expected that such properties will be maintained when it is used to functionalize polysulfone (denoted here mPSF). However, polysulfone has a hydrophobic character (θ > 90°). The influence of H-bonding interactions onto the surface hydrophobicity of the materials was deeply investigated by Nakamura et al. [62]. Based on this study and also on FTIR spectra, the hydrophobicity of the mPSF sample is attributed to the formation of intermolecular hydrogen bonding interactions between OH groups of the PVA and APSF moieties. The surface wettability is known to be strongly affected by the surface chemistry of the carbon-based materials [63,64]. The necessity to improve the interfacial polymer-filler interactions led to the development of several chemical routes of MWCNTs functionalization [65,66]. In this study, the matrix was compatibilized with the filler by its functionalization with hydrophilic OH groups. It is observed that PVA/mMWCNTs composite system presents a gradual decrease in the water contact angle with the increase percent of the filler amount (0.5, 1, and 2.5 wt%),leading to an improvement of the surface hydrophilicity. This behavior could be associated to the increase in surface polarity ($\gamma_{s}^{p}$) as there are fewer polar groups in PVA to interact with those of the modified filler, so more polar groups from mMWCNTs remain available on the composite film surface.

In contrast, the mPSF/mMWCNTs nanocomposites display an easy increase in water contact angle after more filler addition in the system. The new mPSF matrix has more polar groups that can interact with the hydroxyls of the filler. For this reason, upon reinforcement, fewer polar groups remain at the sample surface. This could explain the contact angle data for this composite system that reflect an easy increase in surface hydrophobicity upon mMWCNTs addition up to 2.5 wt%. In the case of all composites containing 5 wt% mMWCNTs, the water contact angle is suddenly increasing. At this composition, the introduced filler particles begin to agglomerate and interact not only by van der Waals forces, but also by means of hydrogen bonding, leaving unavailable polar groups at the sample surface.

The contact angle and wettability analyses provide important information regarding the applicability of materials in surface related applications. According to the work of Khan et al. [67], the problem of hydrophilicity of PVA-based materials for biomedical purposes was analyzed. They show that filtration of biological fluids should have an absorbency level from 70 to 100 water contact angle.

Table 3 illustrates the calculated data regarding the polar and disperse surface tension components of all the studied samples. For the PVA/mMWCNTs composite system, it is observed that the polar component is gradually increasing as the mMWCNTs percent increases (0.5, 1, and 2.5 wt%), whereas the dispersive component decreases. For the other composite system (mPSF/mMWCNTs), the polar component is slightly decreased as compared to the dispersive one. However, for both types of composites containing 5 wt% mMWCNTs, a significant increase in the surface dispersive character as supported by the water contact angle data was observed.

*γ^t^* = total surface free energy,
$\gamma_{sv}^{d}$ = dispersive or non-polar component, $\gamma_{sv}^{p}$ = polar component, *γ_sl_* = solid-liquid interfacial tension, Sc = spreading coefficient.

The hydrophobic/hydrophilic nature of the synthesized materials can be further analyzed by calculating several parameters that are derived from surface tension polar and disperse components (Equations (1)–(3)).

Table 3 contains the calculated values of the $\;\gamma_{sl}$ and S_c_ parameters. As in the case of water contact angle data and surface tension parameters, the interfacial surface tension ranges similarly with reinforcement degree of mMWCNTs. The spreading coefficient ($S_{c}$) describes the mechanisms related to wetting of a liquid when placed on the solid surface of the polymer, reflecting the thermodynamic stability of a liquid layer in contact with the solid material. For positive values of $S_{c}$, the surface presents preponderant adhesive character so the fluid spreading is taking place (complete wetting);conversely for negative values of this parameter, the cohesive properties prevail and partial wetting is noticed. For all samples, the spreading coefficient presents negative values, indicating prevalent cohesion interactions at the interface in detriment to water/sample adhesion interactions.

### 3.4. Adsorption Isotherms

Adsorption/desorption isotherms were registered to obtain information about the water sorption behavior [68] for PVA, mPSF, and their corresponding mMWCNTs composite systems. These results are shown in Figure 6.

According to IUPAC classification [69], the sorption/desorption curves exhibit a type-IV shape for all studied materials. This is specifically for capillary condensation and physical multilayer adsorption [70]. In addition, all of the samples display hysteresis between sorption and desorption (classified as H_2_-typein combination with H_3_-type) [71,72] being characteristic of porous solid structures [73]. The hysteresis of type H_2_ is due to network effects [74] given by the complex pore structures existing in both types of composite samples (PVA/mMWCNTs and mPSF/mMWCNTs) at low values of RH, where a low water sorption is observed. An indicator of the presence of the macropores is given by the shape of the H_3_ hysteresis (up to 30 RH%) observed in all prepared materials. In addition, the interspace between the different mMWCNTs filled into both matrices could be responsible for this H_3_-type of hysteresis. Beyond 30 RH%, one may notice a sharp increase in water sorption, for all nanocomposite systems. Unclosed hysteresis for the first isotherms found for both nanocomposite systems show an irreversible adsorption of water, which is similar to the work reported by Inagaki et al. [75]. As a consequence, the rehydration of the sample surface takes place [75].

Comparison of the DVS data obtained for the pure PVA and mPSF matrices reveal the peculiarities of each polymer structure. In the case of PVA, the main chain is connected to non-bulky side groups that enable a certain degree of intrinsic porosity. The latter is considerable smaller compared to the porosity of mPSF, which has a rigid backbone linked to bulky substituents. The differences in intrinsic porosities are reflected in distinct sorption characteristics of the unfilled and corresponding reinforced samples. It was reported that water absorbed in carbon filled materials follow the Dubinin–Serpinsky (D-S) mechanism [76].

The structure of the material affects the water sorption behavior [77,78]. Table 4 displays the pore diameter and the BET analysis of the nanocomposites materials. Analyzing the results, a gradual increase in the pore size diameter as the mMWCNTs filler amount were added in the PVA matrix (from 0.5 to 2.5 wt%) was observed simultaneously with a sharp decrease in the surface area for all the nanocomposite materials.

The enhancement of pore diameters is attributed to the combined effects given by the nano-confinements and surface functionalization of the MWCNTs [79]. There are some direct strategies to create porosity in the polymer composites. Some of these include utilization of templates, etching and hydrolytic processes, polymeric blending, or introduction/removal of ordered solid compounds intensively applied to obtain tailored pore shapes and dimensions. The “reverse template” approaches led to regular and uniform size pores with possibility to increase the pore diameter by variation in fillers size. Closely interconnected and regular pores in polysulfones have been achieved by introducing different inorganic fillers (salts) into the polymer matrix [80].

However, an in-depth understanding on the dependence of the surface features with the porosity of polysulfones embedding CNTs is still lacking. Reporting data showed that these membranes are useful in selective separation processes of gases, ions, or molecules. The modification of polysulfones induced an increased hydrophilicity and, as a result, an increased antifouling behavior. Some of these membranes are successfully applied in sea-water desalination, the CNTs incorporation proving a better flux, a high mechanical strength, and acid resistance. Moreover, the composites membranes with CNTs showed a high efficiency by phase inversion, being the preferred candidates for pressure retarded systems and osmosis. [81].

In our samples, the decrease in BET area could be explained by the blockage of the pore due to creation of polar OH groups formed on the surface of the MWCNTs material. For the latest sample in the PVA/mMWCNTs composite system, an insignificant decrease in the pore diameter was recorded. A different behavior was evidenced in the case of the samples from the mPSF/mMWCNTs nanocomposite system. The decrease in pore size diameter with the increase in filler amount of mMWCNTs into mPSF matrix associated with the increase in the surface area was observed. However, there was no significant change in pore diameter for both 0.5 and 1 wt% mMWCNTs filler embedding in mPSF matrix.

### 3.5. Morphological Analysis

Changing the mobility of the polymer chain will induce the modification of the structural properties of the final nanocomposite [82]. Figure 7 displays the morphology of the composite systems containing different amounts of mMWCNTs filler in the PVA matrix. A uniform structure is observed for all the PVA/mMWCNTs composite systems. This indicates that MWCNTs are spread homogeneously throughout the entire surface of the material after their functionalization with polar groups (OH). The OH groups created on the surface of MWCNTs interact very well with the OH groups of the PVA matrix (as shown in the FTIR analysis), preventing their agglomeration into the matrix. The dark spots observed on the SEM images are mMWCNTs.

Figure 8 shows the morphology of the composite system containing different amounts of mMWCNTs filler in the mPSF matrix. Similarly, a more dense and compact morphology with a homogenous distribution of mMWCNTs filler amount was shown for the mPSF/mMWCNTs composite system. The pores clearly observed on the surface of the materials indicate a porous material, which is necessary for the targeted application. This is in accordance with the work reported by Wang [83]. The porous structure of this sample was attributed to the compaction of the polymer network during the synthesis process.

### 3.6. In Vitro Hemolysis Assay

Hemocompatibility of the materials constitutes an important feature for their use in blood-contacting applications. The hemolysis assay was performed for the materials containing 1 wt% and 5 wt% mMWCNTsin both the mPSF and PVA matrices. The test results are shown in Figure 9 and were expressed as means ± SD (n = 3). It is observed that all tested materials are non-hemolytic. For PVA/mMWCNTs nanocomposite systems, a degree of hemolysis that varied between 1.2 and 1.75% was observed. These results were attributed to the highly hydrophilic surface of the material, being well correlated with the contact angle measurements discussed above. The mPSF/mMWCNTs nanocomposite system having the highest filler loadings (5 wt%) in matrix showed a slightly hemolytic behavior. These materials had a degree of hemolysis in the range 1.01–2.3%. As expected, PVA-based systems have a lower degree of hemolysis. We can conclude that both systems have a good compatibility with blood.

## 4. Conclusions

New mPSF/mMWCNTs nanocomposite materials designed for water purification or blood-contacting materials were prepared. The characteristics of the prepared materials were compared with those of PVA/mMWCNTs composites. Important structural differences, mainly on O-H vibrations, in the composition of the PVA and mPSF matrices before and after embedding of mMWCNTs fillers in different concentration were evidenced by using the IR spectral subtraction of the spectra. An increased hydrophilicity of the materials was observed after the incorporation of mMWCNTs into PVA and mPSF matrices. The DVS analysis revealed that the water sorption behavior of the studied systems is affected by the structure of the material. All of the systems presented a type-IV isotherm with hysteresis between sorption and desorption (classified as H2-type in combination with H3-type) in the case of both nanocomposite materials. SEM images showed more dense and compact morphology with homogenous distribution of mMWCNTs filler amounts into the mPSF matrix as compared with PVA/mMWCNTs materials. The presence of pores onto the surface of the mPSF material indicated porous material, which is necessary for the pursued applications. Hemolysis results showed that all tested materials are non-hemolytic, concluding that both composite systems have a good compatibility with blood.

## Data Availability

The data presented in this study are available on request from the corresponding author.

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
