# Peer review of "MWCNTs Composites-Based on New Chemically Modified Polysulfone Matrix for Biomedical Applications"

_nanomaterials, 2022, doi:10.3390/nano12091502_

Round 1
Reviewer 1 Report
The paper entitled “MWCNTs composites-based on new chemically modified polysulfone matrix for biomedical applications” deals with the study of PSF surface modification by covalent link with PVA and subsequently the incorporation of mMWCNTs. The PSF functionalized membrane was then compared with the composite PVA/MWCNTs. The surface modification was determined by ATR-FTIR and correlated with the wettability study to determine the impact of the modification in the physical properties. Finally, hemolysis assay was performed and showed promising success for blood filtration.
I recommend major revisions.
1/ The use of PVA is not clear in the abstract. The authors are suggested to clarify the use of PVA to modified the PSF prior to incorporation of mMWCNTs. PVA must be mentioned in the keywords.
2/ The authors are suggested to develop the state of the art regarding the use of mMWCNTs for blood-device. Which advantages should we expect?
3/ The protocol of the polysulfone chloromethylation is missing, even though it was earlier published, I recommend to specify the synthesis in the experimental part.
4/ A scheme to explain the different interactions within the composite would be highly appreciable.
5/ There was a confusion with the IR figures. Figure 3c and d are the same and the figure 4 must be incorporated with the figure 3.
6/ The figure 5 is called Figs 11 &12 in the text (p11 l.259; p12 l.264)
7/ How many times the wettability studies have been performed? The authors are suggested to precise the standard deviation and the reproducibility of these measures.
8/ The increase of water contact angle for the composite 5% for the membrane mPSF follows the trend but is not really remarkably increasing, unlike to what is observed for the PVA/mMWCNTs. Therefore, I would suggest to dissociate the behavior discussion for both membranes. Hence, why such effect is not better visible for the mPSF?
9/ The authors are suggested to present the equations (3,4,5, and 6) in the experimental section.
10/ Could you comment on the stability of the composite under fluid flow, especially regarding the release of mMWCNTs?
11/ The discussion regarding the creation of porosity for the polymer composite could be developed (pp20 l.425).
Reviewer 2 Report
The paper "MWCNTs composites-based on new chemically modified polysulfone matrix for biomedical applications" are presented interesting and new results on new composite materials consisting of modified polysulfone (mPSF) and functionalized multiwall carbon nanotubes (mMWCNTs). Paper can be published in Nanomaterials mdpi after major revision. The following points should be mentioned.
1. It will be useful to add a Scheme that will illustrate all stages of the synthesis of the composite materials. It is difficult for understanding what types of samples were synthesized.
2. Why as the final product was chosen modified polysulfone with PVA and carbon nanotubes. Appropriate discussion should be provided. If I understood rightly, polysulfone membranes were not studied but only modified polysulfone, why?
3. Please add a Table where will be suggested ATR-FTIR picks and groups assigned to these picks. Information in figures is too much to understand adequately.
4. What about errors for contact angle values. Why dependence between filer amount and wetting contact angles for PVA/mMWCNTs composite system is another than for mPSF/mMWCNTs composite system (I suggest on 2.5 % filler amount for PVA).
6. I suggest to cite relevant publications where similar results were presented.
https://doi.org/10.1016/j.apsadv.2021.100104
https://doi.org/10.1007/s10965-020-02335-7
https://doi.org/10.1016/j.cis.2013.05.005
Round 2
Reviewer 1 Report
The authors have well replied to the comments, and the revised version of the paper can be published.
Reviewer 2 Report
How can I have a comment on the author's response such as downloading the manuscript I see only the old version, the quality of the manuscript was essentially improved.